# Differences in Antioxidant Potential of *Allium cepa* Husk of Red, Yellow, and White Varieties

**DOI:** 10.3390/antiox11071243

**Published:** 2022-06-24

**Authors:** Irina Chernukha, Nadezhda Kupaeva, Elena Kotenkova, Daniil Khvostov

**Affiliations:** 1V. M. Gorbatov Federal Research Center for Food Systems, Experimental Clinic and Research Laboratory for Bioactive Substances of Animal Origin, Talalikhina St., 26, 109316 Moscow, Russia; imcher@inbox.ru (I.C.); nvkupaeva@yandex.ru (N.K.); 2V. M. Gorbatov Federal Research Centre for Food Systems of RAS, Laboratory of Molecular Biology and Bioinformatics, Talalikhina St., 26, 109316 Moscow, Russia; d.hvostov@fncps.ru

**Keywords:** *Allium cepa*, onion, antioxidant, phenolic compounds, total antioxidant capacity, UPLC, chemiluminescence, quercetin

## Abstract

The effective management of agro-industry organic waste for developing high-commercial-value products is a promising facet of the circular economy. Annually, more than 550,000 tons of waste that is potentially rich in biologically active substances is generated worldwide while processing onions (*Allium cepa* L.). The antioxidant potential of red, yellow, and white onion husks was studied using FRAP, ORAC, chemiluminescence, and UPLC-ESI-Q-TOF-MS analysis methods. The extraction of phenolic compounds from onion husks was more effective when using an aqueous solution of 70% ethanol as compared with water. Ethanolic extract from red onion husks exhibited the highest TAC_ORAC_ and TAC_FRAP_ values, averaging 2017.34 µmol-equiv. Trolox/g raw material and 2050.23 µmol-equiv. DQ/g raw material, respectively, while the white onion exhibited much lower levels of antioxidants. According to the chemiluminescence results, it was determined that the red and yellow onion husks contained antioxidants of three types of power, while white onion husks only contained medium and weak types. The highest content of flavonoids was found in red onion husks, averaging 1915.90 ± 9.92 µg-eq. DQ/g of raw material and 321.42 ± 2.61 µg-eq. DQ/g of raw material for ethanol and water, respectively, while yellow onion husks exhibited 544.06 ± 2.73 µg-eq. DQ/g of raw material and 89.41 ± 2.08 for ethanol and water, respectively. Quercetin and its glycosides were the most representative flavonoids, and a number of substances with different pharmacological and biological properties were also identified.

## 1. Introduction

At present, the circular economy (CE) is a popular framework, and is promoted by many industries around the world [1]. The linear economy is based on the “take, make and dispose” principle [2]; the CE, in contrast, is aimed at the efficient use of resources through minimizing waste, and the retention of long-term value [3]. Therefore, the CE is a potential solution that promotes the efficient use of resources in order to maximize economic benefits while alleviating pressure on the environment [4]. The European Commission determined that the transition to the CE could provide an annual economic benefit of EUR 600 billion for the manufacturing sector alone [1]. The agro-industry produces a large amount of organic waste that often ends up in landfills or is used to produce products with low added value; however, the CE concept aims at converting this waste into products with high commercial value, such as medicines, nutraceuticals, and cosmeceuticals [5,6]. Secondary products of the food industry often contain valuable molecules that can be used as functional ingredients in the food, cosmetics, and pharmaceutical industries [7].

Onion (*Allium cepa L.*) is the second most cultivated crop worldwide after tomatoes, with a harvest estimated to be approximately 89 million tons [5,6]. The huge popularity of the onion is related to its versatility. It is used as a raw food and in different cooked forms, e.g., baked, boiled, braised, grilled, fried, etc. [8]. In recent years, global onion production has increased by at least 25% [7,9], which is linked not only to the use of onions as a flavored vegetable or spicy ingredient, but also to the use of onions as a source of bioactive phytonutrients [9,10]. It is known that onions are rich in antioxidant compounds, and that their consumption contributes to the prevention of certain diseases associated with oxidative stress, and numerous epidemiological studies have confirmed that regular onion consumption reduces the incidence of various forms of cancer, along with cardiovascular and neurodegenerative diseases [8,10].

More than 550,000 tons of bio-waste are generated during onion processing [5,6], which consists of peeling, slicing, and dicing, and these operations generate a large amount of waste [11]. Interest in onion waste has grown significantly in recent years, as evidenced by the increase in the number of related scientific studies in the past 2 years. This is due to the cheapness and availability of this waste. The waste biomass mainly consists of skin/peel/husk (the outermost layers), roots, tops of the bulbs, and deteriorated bulbs [6,12]; however, onion skin represents the main waste component in onion processing (up to 60%) [7,11]. Onion waste is characterized by an unpleasant taste and smell; therefore, it is not suitable for animal feeding or as organic fertilizer, and is usually sent to landfill, causing an environmental problem [5,6,7]. On the other hand, onion peel is rich in polyphenolic antioxidants—mainly quercetin and its derivatives; glucosides, which belong to the flavonoids group; and ferulic acid, gallic acid, and kaempferol, which demonstrate significant beneficial effects associated with various biological activities, including antidiabetic, antioxidant, anti-inflammatory, antitumor, antimicrobial, and enzyme-inhibitory effects [5,13]. Concerning onion varieties, quercetin and its derivatives, along with glucosides, are the predominant flavonols in all types of onions, regardless of the white, yellow, or red color, while anthocyanins are mainly present in red onions, wherein they make up approximately 10% of the total flavonoid content in fresh weight [7,9]. Contents of prominent flavonoids such as quercetin and its glucosides are the highest in pearls, followed by red, yellow, and white onion husks [11,14], and strongly depend on extraction conditions [15].

Onion production in Russia represents a significant portion of total vegetable production. White, yellow, and red onions are consumed, with yellow being the predominant type. The aim of this study was to assess the antioxidant potential of husk waste obtained from these three varieties of onions.

## 2. Materials and Methods

### 2.1. Preparation of Onion Husk Extracts

For the preparation of husk extracts, 3 varieties of onions were purchased in the spring of 2021 at the “Perekrestok” supermarket in Moscow, Russia. Yellow, red, and white onions with an even, rounded shape were selected for research. The husk was ground to a size of no more than 5 mm. The husks were soaked in 70% ethanol (1 g/15 mL) for 24 h with gentle shaking at room temperature, and then in water (1 g/15 mL) at an initial temperature of ≈98 °C for 15 min with gentle shaking. The content was filtered through a filter paper and stored in airtight bottles in a refrigerator at 4 °C until use.

### 2.2. Chemicals

Fluorescein sodium salt (purity ≥ 97%, Saint Louis, MO, USA), 2,2′-azobis (2-methylpropionamidine) dihydrochloride (AAPH, purity ≥ 97%, Saint Louis, MO, USA), quercetin (purity ≥ 95%, Bangalore, Karnataka, India), (±)-6-hydroxy-2,5,7,8-tetramethyl-chromane-2-carboxylic acid (Trolox, purity ≥ 97%, Schaffhausen, Switzerland), iron(III) chloride hexahydrate (purity ≥ 99%, Taufkirchen, Germany), luminol (purity ≥ 97%, Saint Louis, MO, USA), (+)-sodium L-ascorbate (purity ≥ 98.5%, Wuxi, China), standard (purity > 98%, Darmstadt, Germany) flavonoids including quercetin, and acetonitrile (purity ≥ 99.9%, Darmstadt, Germany) were purchased from Sigma-Aldrich (Saint Louis, MO, USA).

Dipotassium hydrogen phosphate anhydrous (purity ≥ 98%), potassium dihydrogen phosphate (purity ≥ 98%), sodium acetate anhydrous (purity ≥ 99%), hydrogen peroxide (H_2_O_2_, purity ≥ 33%), DL-α-tocopherol (purity ≥ 99%), peroxidase from horseradish (M ≈ 40,000 g/mol, activity: 235.9 U/mg), hydrochloric acid (HCl, 37%), and formic acid (FA, purity ≥ 98%) were purchased from PanReac AppliChem (Darmstadt, Germany).

Acetic acid (purity ≥ 99.8%) was purchased from Component-Reaktiv (Moscow, Russia). 2,4,4-Tris(2-pyridyl)-1,3,5-triazine (TPTZ, purity ≥ 99%) was purchased from Thermo Scientific Acros (Hong Kong, China).

Deionized water for chromatography (18 Ω) was obtained using a Milli-Q Merck water purification system (Millipore, Darmstadt, Germany).

### 2.3. UPLC-ESI-Q-TOF-MS Analysis

Chromatographic separation of the studied polyphenol compounds was performed using an UHPLC 1290 Infinity system (Agilent Technologies, Santa Clara, CA, USA) [16]. Separation was achieved using a ZORBAX RRHD Eclipse Plus C18 analytical column (2.1 mm × 100 mm, 1.8 μm particle size, Agilent Technologies, Santa Clara, CA, USA). The mobile phase—H_2_O (A) and ACN (B) prepared with 0.1% formic acid (Sigma-Aldrich, Darmstadt, Germany) *v*/*v*—was pumped at a flow rate of 0.4 mL/min, whereas the injection volume was 3 μL. Chromatography was carried out in a linear gradient as follows: 15% solvent B for 2 min, from 15% to 25% solvent B for 3 min, from 25% to 32% solvent B for 1 min, from 32% to 40% solvent B for 5 min, from 40% to 55% solvent B for 4 min, from 55% to 95% solvent B for 4.50 min, and 95% solvent B for 1.50 min. Thereafter, the gradient switched back, returning to the initial 100% A in 2 min. The total analysis time was 25 min.

A time-of-flight mass spectrometric detector—i.e., Agilent 6545XT AdvanceBio LC/Q-TOF (Agilent Technologies, Santa Clara, CA, USA), equipped with DuoJet Stream ESI (Agilent Technologies, Santa Clara, CA, USA) and an ion funnel (Agilent Technologies, Waldbronn, Germany) set in negative ionization mode—was coupled with the UHPLC system. The main operating parameters of the Q-TOF MS were set as follows: capillary voltage 3500 V; nozzle voltage 1000 V; drying gas flow 12 L/min at 300 °C; gas flow through the casing 11 L/min at 300 °C; and atomizer pressure 35 psi. The high-pressure ion funnel was operated at a high frequency (RF) of 150 V, the low-pressure funnel at 65 V RF, and the octopole at 750 V. Analyses were performed in full-scan MS mode and in auto MS/MS mode, with full scanning from 50 to 1700 *m*/*z*. Fragment spectra were obtained for each mass using the collision energy set at 20 eV. In the MS/MS experiments, nitrogen was used as the collision gas. A reference solution was used—i.e., purine ([M + H]^+^ = 121.0509) and Agilent compound HP0921 ([M + H]^+^ = 922.0098)—to calibrate the internal mass throughout the analysis. Detected compounds were identified by MS fragmentation using the MSDIAL software (ver. 4.60, RIKEN CSRS, Yokohama City, Japan) [17]. The measurements were carried out in triplicate. Flavonoid contents were determined according to a standard curve using dihydroquercetin (DQ) in the concentration range of 1–1000 ng/mL. Values for each flavonoid are expressed in µg-eq. DQ/g of raw material.

### 2.4. Determination of Total Antioxidant Capacity (TAC)

#### 2.4.1. Ferric Reducing Antioxidant Power (FRAP) Assay

Determination of TAC using the FRAP method was performed on an SF-2000 spectrophotometer (OCB «Spectr», St. Petersburg, Russia) using the method of Benzie and Strain (1996) [18], with the author’s modification. The fresh FRAP solution was prepared by mixing 300 mM acetate buffer (pH = 3.6), 10 mM TPTZ (prepared in 40 mM HCl), and 20 mM ferric (III) chloride aqueous solution at a ratio of 10:1:1 (*v*/*v*/*v*). Volumes of 1.45 mL of FRAP reagent and 50 μL of the sample, the standard, or distilled water for measuring the control sample were added to the tube. The reaction mixture was incubated for 30 min at 37 °C in the dark. The optical density was measured at a wavelength of 594 nm. TAC was determined according to a standard curve using dihydroquercetin (DQ) in the concentration range of 15–100 μM. Depending on their activity, the extracts were diluted with distilled water. The TAC of the onion husk extracts is expressed in µmol-equiv. DQ/g raw material.

#### 2.4.2. Oxygen Radical Absorbance Capacity (ORAC) Assay

Determination of TAC using the ORAC method was performed on a Fluoroskan Ascent FL system (TermoLabsystems, Vantaa, Finland) using black 96-well plates [19], with the author’s modification. A total of 30 µL of sample or standard and 200 µL of 0.5 µM sodium fluorescein were added to the wells, and the microplates were covered with film (SSIbio, Lodi, CA, USA) and placed into the Fluoroskan Ascent FL for 30 min at 37 °C. Then, 30 µL of 153 µM AAPH was added to each well, and the fluorescence was measured at 37 °C for 60 min at 5 min intervals. The excitation wavelength was 485 nm, and the emission wavelength was 535 nm. The TAC of each sample was determined four times. TAC was determined according to a standard Trolox curve in the concentration range 5–75 µM. Depending on their activity, the samples were diluted with 75 mM phosphate buffer (pH = 7.4). TAC is expressed in µmol equiv. Trolox/g raw material.

### 2.5. Chemiluminescence Assay 

Screening of the complex composition of onion husks by studying the type of antioxidant interaction with free radicals in plant extracts was carried out by analyzing the effects of such processes on the kinetic chemiluminescence curves, in accordance with the methodology in [20] using the Lum-100 chemiluminometer (DISoft, Moscow, Russia) and the PowerGraph 3.3. software (DISoft, Moscow, Russia). In total, 40 µL of 1 mM luminol solution, 20 µL of 0.5 mM horseradish peroxidase solution, 10 µL of extract or standard AO in various dilutions, and 930 µL of 20 mM phosphate buffer (pH = 7.4) were added to the glass tube. The glass tube was placed in the chemiluminometer and a background signal was observed for 48 s, and then 100 µL of 1 mM hydrogen peroxide solution was added to the reaction mixture, and the chemiluminescent signal was recorded for 10 min. The types of free radicals scavenged by antioxidants in the extracts were determined by comparing the obtained kinetic curves for several dilutions of the sample with the curves of standard antioxidants. Quercetin, tocopherol, and sodium ascorbate were used as standard AOs.

### 2.6. Statistical Analyses

The STATISTICA 17.0 software was used in this study for the statistical analysis. The results were calculated as mean ± SD. Significant differences were calculated by one-way ANOVA followed by Tukey’s HSD test. Differences with *p*-values < 0.05 were considered statistically significant. After UPLC-ESI-Q-TOF-MS data processing—which included peak collection, deconvolution, compound identification, and peak alignment—a multivariate analysis was performed using principal component analysis (PCA) in the MS-DIAL software (ver. 4.60, RIKEN CSRS, Yokohama City, Japan) [17].

## 3. Results

### 3.1. Identification of Active Compounds in the Extract of Onion Peels

Aqueous and alcoholic extracts from various types of onions were analyzed using UPLC-ESI-Q-TOF-MS analysis. More than 100 compounds were obtained using the MSDIAL accurate mass tolerance MS1—0.01 Da and MS2—0.05 Da program parameters. A total of 30 compounds was manually selected, including organic (n = 3), organosulfur (n = 1), and phenolic acids (n = 6), 1,4-benzodiazepines (n = 2), flavonoids (n = 10), and other lesser studied polyphenolic compounds (n = 8). Appendix A shows the mass parameters and identification characteristics of all manually selected compounds (score ≥ 80%).

Quantitative determination of the main flavonoids in ethanolic and water-based extracts of onion husks was carried out using calibration curves of quercetin; the regression coefficient was >0.990. Table 1 shows the main compounds determined in water and ethanolic extracts prepared from red, yellow, and white onion husks.

Quercetin and spiraeoside were the most representative flavonoids observed in onion husk extracts. The quercetin content in the ethanolic extract from red onion husks was the highest, and exceeded values from the water group and the ethanolic extract from yellow onion husks by 12-fold (*p* < 0.05) and 3.2-fold (*p* < 0.05), respectively. The content of spiraeoside in the ethanolic extract from red onion husks was the highest, and exceeded values from the water group and the ethanolic extract from yellow onion husks by 4.1-fold (*p* < 0.05) and 4.3-fold (*p* < 0.05), respectively. Polyphenolic compounds, due to their lower polarity, were more easily extracted in an ethanolic solution than in water (Figure 1) for both red and yellow husks, while in both the ethanolic and water extracts from the white onion husks, flavonoids were not detected. The total flavonoid content in the ethanolic extract from red onion husks was the highest, and exceeded values from the water group, the ethanolic group, and the water extract from yellow onion husks by 6.0-fold (*p* < 0.05), 3.5-fold (*p* < 0.05), and 21.4-fold (*p* < 0.05), respectively.

### 3.2. Total Antioxidant Capacity Analysis

In onion husk extracts, the TAC values were determined using the ORAC and FRAP methods. The ORAC fluorescent method was used to evaluate the contribution of antioxidants demonstrating an HAT (hydrogen atom transfer) mechanism of action, and describing the ability of antioxidants to neutralize free radicals by hydrogen donation. The FRAP photometric method was used to evaluate the contribution of antioxidants demonstrating an SET (single-electron transfer) mechanism of action, and describing the ability of antioxidants to interact with free radicals by transferring a single electron [21]. The use of these two methods made it possible to study the contribution to the antioxidant potential of samples of both antioxidants acting through the HAT mechanism and those acting through the SET mechanism. The obtained TAC values for all extracts are shown in Table 2.

The TAC_ORAC_ and TAC_FRAP_ values varied significantly depending on the type of onion for both ethanolic and aqueous extracts. The ethanol extract from the red onion demonstrated the highest TAC value, while the TAC values of white onions were extremely low. The TAC_ORAC_ value of the ethanol extract from the red onion husk exceeded the TAC_ORAC_ values of similar extracts from yellow and white onions by 2.19-fold (*p* < 0.001) and 122.41-fold (*p* < 0.001), respectively, and the TAC_FRAP_ values of the ethanol extract from the red onion husk exceeded the TAC_FRAP_ values of similar extracts from yellow and white onions by 2.71-fold (*p* < 0.001) and 1009.97-fold (*p* < 0.001), respectively. A similar tendency was observed for aqueous extracts from onion husks. The TAC_ORAC_ of the water extract from the red onion husk was the highest, and exceeded the TAC_ORAC_ values of similar extracts from yellow and white onions by 1.58-fold (*p* < 0.001) and 104.86-fold (*p* < 0.001), respectively, while the TAC_FRAP_ values of the water extract from the red onion husk exceeded values of similar extracts from yellow and white onions by 2.51-fold (*p* < 0.001) and 259.77-fold (*p* < 0.001), respectively. 

The TAC values of ethanol extracts from red and yellow onion husks were statistically higher than those from water. The TAC_ORAC_ values of ethanol extracts from red and yellow onion husks exceeded the TAC_ORAC_ values of the water extracts by 3.74-fold (*p* < 0.001) and 2.69-fold (*p* < 0.001), respectively, while the TAC_FRAP_ values exceeded those from water by 2.89-fold (*p* < 0.001) and 2.68-fold (*p* < 0.001), respectively. This observation suggests that the husks of red and yellow onions contain significantly more fat-soluble antioxidants than water-soluble antioxidants, acting through both the HAT mechanism and the SET mechanism. No statistically significant differences were found for white onion extracts, but the TAC_ORAC_ value of the ethanol extract was 3.2-fold higher than that of the water group.

When comparing the TAC values obtained by different methods for the same extracts, statistically significant differences were only observed for extracts from the yellow onion husk and the water extract from the red onion husk. The TAC_ORAC_ values of the ethanol and water extracts from the yellow onion husk exceeded the TAC_FRAP_ values by 1.22-fold (*p* < 0.001) and 1.21-fold (*p* < 0.001), respectively. The TAC_FRAP_ value of the water extract from the red onion husk exceeded the TAC_ORAC_ value by 1.31-fold (*p* < 0.001). The obtained data indicate that the antioxidant potential of the yellow onion husk is more vigorously contributed to by AOs acting through the HAT mechanism, regardless of the solvent. Concerning the red onion husk, the highest TAC was observed for the ethanol extraction, the contribution to which was equal for the AOs acting through the HAT and the SET mechanisms, while for the aqueous extraction, an increase in the contribution of AOs acting through the SET mechanism was observed.

### 3.3. Chemiluminescence Analysis

The chemiluminescence method was used to investigate the interaction type of standard AOs with FR by obtaining the kinetic curves of the change in the intensity of CL over time. To study the antioxidant activity of sodium ascorbate, tocopherol, and quercetin, kinetic chemiluminescence curves were obtained for at least six concentrations prepared from a stock solution. The resulting graphs are shown in Figure 2.

During the kinetic curve study, it was found that AOs, depending on their activity, have at least three types of interaction with oxygen-containing free radicals, which is also consistent with the data presented in the work of G.K. Vladimirov et al. [20].

When studying the effect of sodium ascorbate (Figure 2a) on the kinetics of CL, a latent period was established immediately after the addition of the AO, which was revealed by an almost complete suppression of the intensity of CL. This duration was proportional to the concentration of this type of AO. Moreover, invariance of the slope of the CL curves and the intensity of CL on the plateau was noted, i.e., being a strong AO, sodium ascorbate is able to neutralize all radicals, including luminol radicals. The development of CL in this case began only after the oxidation of all ascorbate molecules.

The study of the effect of tocopherol (Figure 2b) on the kinetics of CL demonstrated that the interaction of this AO with CP is characterized by a decrease in the intensity of CL on the plateau. Such a smooth extinguishing of CP is characteristic of weak antioxidants, despite the fact that tocopherol is considered to be one of the most powerful AOs. In [20], it was assumed that such a discrepancy in the antioxidant activity of tocopherol is due to the fact that the FR in this system were in a water medium, while it is customary to study the properties of tocopherol in non-polar media.

The analysis of the obtained kinetic curves of CL for a system with quercetin (Figure 2c) did not reveal a latent period or a significant decrease in the intensity of CL on the plateau. It was found that the main effect of quercetin in this system is characterized by a change in the slope of the curves—i.e., the rate of development of CL—with a slight decrease in intensity on the plateau. This type of interaction between AO and FR is typical of medium-strength AOs. It was also found that the slope increased with a decrease in the concentration of quercetin.

The effects of aqueous and ethanolic extracts from yellow, red, and white onion husks on the kinetics of chemiluminescence in various dilutions were studied. The obtained kinetic curves are shown in Figure 3.

When analyzing the kinetic curves of CL systems with the addition of ethanol (Figure 3a) and water (Figure 3b) extracts from the yellow onion husk, changes in the characteristics of all three types of AO were observed. However, it was noted that, for the ethanol extract, the latent period (strong AO) was longer than for the water extract, whereas the decrease in the intensity of CL on the plateau (weak AO) was more pronounced for the aqueous extract. The graphs for the ethanol (Figure 3c) and aqueous (Figure 3d) extracts from the red onion husk show that the samples also contain three types of antioxidants. It was noted that the duration of the latent period of the ethanol extract from the red onion husk was similar to the duration for the yellow onion, and less than that of the water extract from the red onion husk. In addition, the water extract from the red onion (Figure 3d) exhibited a more pronounced decrease in the intensity of CL on the plateau than the ethanol extract. When analyzing the kinetic curves for the ethanolic (Figure 3e) and water (Figure 3f) extracts from the white onion husk, a change in the slope of the curve corresponding to the presence of medium AO and a decrease in the intensity of CL on the plateau (weak AO) were detected.

It was noted that for ethanol extracts from yellow onion husks (Figure 3a) and red onion husks (Figure 3c), the dilutions were much greater than in the case of aqueous extracts, whereas for white onion husks the difference was insignificant. Such an observation indicates a greater amount of AOs in ethanol extracts from yellow and red onion husks as compared with water extracts, whereas in the case of the white onion husk the amount of AOs in the extracts differed slightly, which correlates with the TAC values obtained using the FRAP and ORAC methods. It is important to note that, in the case of the yellow and red onion husk extracts, the decrease in the intensity of CL on the plateau was similar to the results obtained for quercetin; however, even with the smallest dilution of extracts, the intensity of CL did not reach blank values, and was significantly lower, indicating the presence of weak AOs, but in small quantities. Thus, using the CL method, it was found that the contribution to the TAC value of the red and yellow onion husk extracts is provided by three types of AO, whereas white onion husk extracts contain only medium and weak AOs. Furthermore, the CL method made it possible to establish that alcoholic extracts from red and yellow onions contain more AOs than aqueous extracts, and there were no significant differences in the amounts of AOs in the extracts from the white onion husks.

## 4. Discussion

The following are the generally utilized solvents for extracting polyphenols: methanol, water, chloroform, n-hexane, ethanol, propanol, ethyl acetate, and acetone, along with their water mixtures, with or without acid [22]. Polar solvents are frequently used for the isolation of polyphenols from plants. The most suitable solvents are aqueous mixtures containing ethanol, methanol, acetone, and ethyl acetate [23]. Various studies have revealed that ethanol/water solvents are more effective in extracting phenolic compounds than water, and that ethanol extracts exhibit a higher antioxidant activity than aqueous extracts [24]. Moreover, non-toxic and biodegradable alternatives, such as ethanol, are being explored to some extent as extraction methods to reduce the impact of organic solvents on the environment, while providing similar or even superior performance [25]. Ethanol–water mixtures are recommended for the preparation of plant extracts due to their acceptability for human consumption [26]. In general, 50–80% aqueous ethanol solutions are often used for the extraction of phenolic compounds from different parts of plants [24,26,27,28]. In our study, we revealed that the extraction of phenolic compounds from onion husks was more effective when using 70% ethanol solution as compared with water. Lee et al. reported that ethanol extraction increased the total phenolic and flavonoid contents in the onion peel extract [29]. Viera et al. studied the influence of different concentrations of ethanol (20%, 40%, 60%, and 80%) on the extraction of antioxidants from red onion skin extract; the results demonstrated that a concentration of 80% ethanol was most favorable for the extraction of phenolic compounds, flavonoids, and total anthocyanins, and produced the highest antioxidant activities found using the different methods [30]. However, other researchers have reported that onion peel extracted with a lower concentration of ethanol (50%) exhibited higher extraction yields and total phenolic contents than extracts from distilled water, or from 70% or 95% ethanol [31]. Other researchers reported that the optimal conditions for quercetin extraction from red onion skin were using 80% ethanol adjusted to pH 1.0 [32].

Quercetin and its glucosides are the most representative flavonoids in onion husks [14]. Quercetin—a plant pigment and, more specifically, a flavonol—is a potent antioxidant flavonoid, and is known to possess protective abilities against tissue injury induced by various drug toxicities. Furthermore, it is thought to exert many beneficial health effects, including protection against various diseases such as osteoporosis, lung cancer, and cardiovascular disease [33]. Kwak et al. reported that quercetin was the predominant compound in red onion, while in yellow onion quercetin 3-glucoside levels were much higher, followed by quercetin [34]. However, in our study, quercetin was the predominant compound observed in both red and yellow onion husks. Nile et al. showed that ethanol is the most appropriate solvent for spiraeoside extraction from red onion skin waste [35]. Moreover, spiraeoside, which exhibited promising anticancer effects against HeLa cells, was able to promote apoptosis by activating the expression of caspase-3 and caspase-9 [36]. Patil et al. showed that, in all onions studied, spiraeoside was the main quercetin-containing compound present [37], which is consistent with our data. Hyperoside (quercetin-3-O-D-galactoside) has different pharmacological actions, such as anti-inflammatory, antidepressant, neuroprotective, cardioprotective, antidiabetic, anticancer, antifungal, radioprotective, gastroprotective, and antioxidant activities [38]. Interestingly, it was mainly detected in the ethanol extract from red onion husk, while in the other extracts it was detected in very small amounts, or was not detected at all. 

Myricetin is a common plant-derived flavonoid that exhibits a wide range of activities, including strong antioxidant, anticancer, antidiabetic, and anti-inflammatory activities. Furthermore, it may protect against Parkinson’s and Alzheimer’s diseases [39]. Benito-Román et al. showed that red onion skin waste contains more myricetin than that of yellow onions [6], as was the case in our results, although the difference in content was not as distinct. Laricitrin, a 3′-MeO analog of myricetin [40], is a less common flavonol that has not been thoroughly studied as compared to kaempferol or quercetin [41]. It was reported that laricitrin could suppress certain factors and decrease the progression of lung cancer cells [42]. In our study, we did not observe a strong dependence regarding laricitrin extractivity on the type of solvent for red and yellow onion husks. Taxifolin, a unique bioactive flavonoid, is a powerful antioxidant with a well-documented effect in the prevention of several malignancies in humans, along with activity against inflammation, microbial infection, oxidative stress, cardiovascular disease, and liver disease [43]. Taxifolin naturally occurs in onions [44], and our data showed that the taxifolin content was higher in red onion husks as compared with those of yellow onions. Tectorigenin has been reported to possess antioxidant, hair-darkening, and anti-inflammatory activities [45], and to have poor water-solubility [46]; therefore, it was detected only in ethanol extracts.

The antioxidant content and TAC values significantly differed between different onion varieties [36]. Red onion husks had the highest contents of polyphenols, followed by the yellow onion husks, with the white onion containing the lowest amounts [36,47,48], but in our study no compounds were detected in the white onion husk extracts. It was reported that white onions contained trace amounts of total quercetin [37], but it was detected in husks, albeit in much lower concentrations than in red and yellow ones [11,14]. Quercetin and myricetin, found in large quantities in red and yellow onion husk extracts, are the main representatives of flavonols, which are considered to be the most abundant type of flavonoids in foods. However, the concentrations of flavonols in different fruits (even those from the same species) vary due to the variation in their biosynthesis in the presence of sunlight [49]. Moreover, the antioxidant content and TAC values of onions significantly depend on the growing region [14]. Although polyphenols exist in several plant materials, their quantity and type are dependent on the extraction methods used, their chemical nature, the particle size, the presence of interfering compounds, and storage conditions [50]. In addition, methanol has been generally found to be more efficient in extraction of lower-molecular-weight polyphenols [23], while we used 70% ethanol solution and water. In summary, we obtained the same pattern for both total flavonoid contents and TAC values: red > yellow > white. Despite there being no compounds detected in the UPLC-ESI-Q-TOF-MS analysis of the white onion husk extracts due to their extremely low concentrations in the prepared extract, which corresponded with low TAC values, the chemiluminescence method allowed us to evaluate the approximate ratio of antioxidants and classify them according to power.

## 5. Conclusions

Onion husk is a good source of antioxidants. Flavonols, flavanonols, flavonoid-O-glycosides, and isoflavones were detected in both the ethanol and water extracts from red and yellow onion husks, but not in the white onion husks. Quercetin and its glucosides were the most representative flavonoids in the onion husks. The highest content of flavonoids was determined in red onion husks, averaging 1915.90 ± 9.92 µg-eq. DQ/g of raw material and 321.42 ± 2.61 µg-eq. DQ/g of raw material for ethanol and water, respectively. In the yellow onion husk, the content of flavonoids was 544.06 ± 2.73 and 89.41 ± 2.08 µg-eq. DQ/g of raw material for ethanol and water, respectively. The results from studying the antioxidant potential of the extracts of three types of onion husks revealed that the ethanol extract from the red onion husk had the highest TAC_ORAC_ and TAC_FRAP_ values, averaging 2017.34 µmol-equiv. Trolox/g raw material and 2050.23 µmol-equiv. DQ/g raw material, respectively. The white onion exhibited much lower levels of antioxidants. It was shown that the husks of red and yellow onions contain significantly more fat-soluble antioxidants and, in the case of yellow onions, the greatest contribution to the antioxidant potential is provided by antioxidants acting through the HAT mechanism. The chemiluminescence results indicated greater amounts of antioxidants in the ethanol extracts from the yellow and red onion husks as compared with the water extracts, whereas in the case of white onion husk, the amounts of antioxidants in the extracts differed slightly, which is consistent with the TAC values obtained using the FRAP and ORAC methods.

## Figures and Tables

**Figure 1 antioxidants-11-01243-f001:**
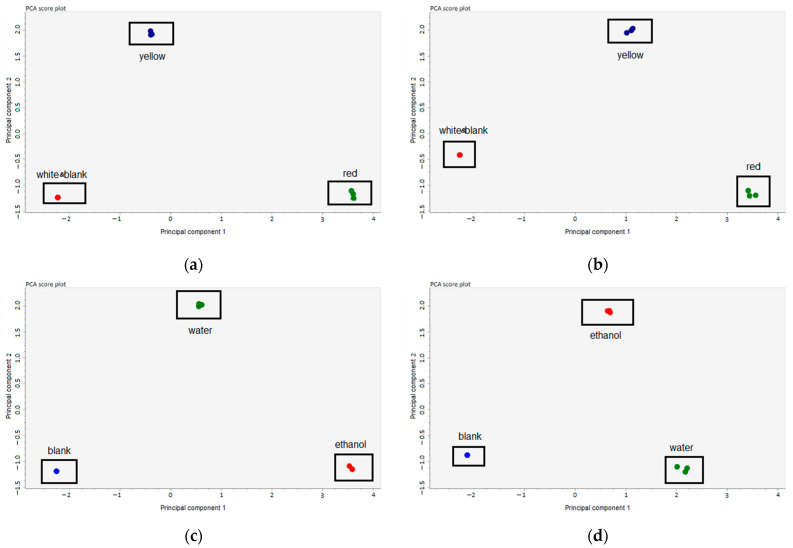
PCA analysis results: (**a**) ethanolic extracts from red, yellow, and white onion husks; (**b**) water extracts from red, yellow, and white onion husks; (**c**) water and ethanolic extracts from red onion husk; (**d**) water and ethanolic extracts from yellow onion husk.

**Figure 2 antioxidants-11-01243-f002:**
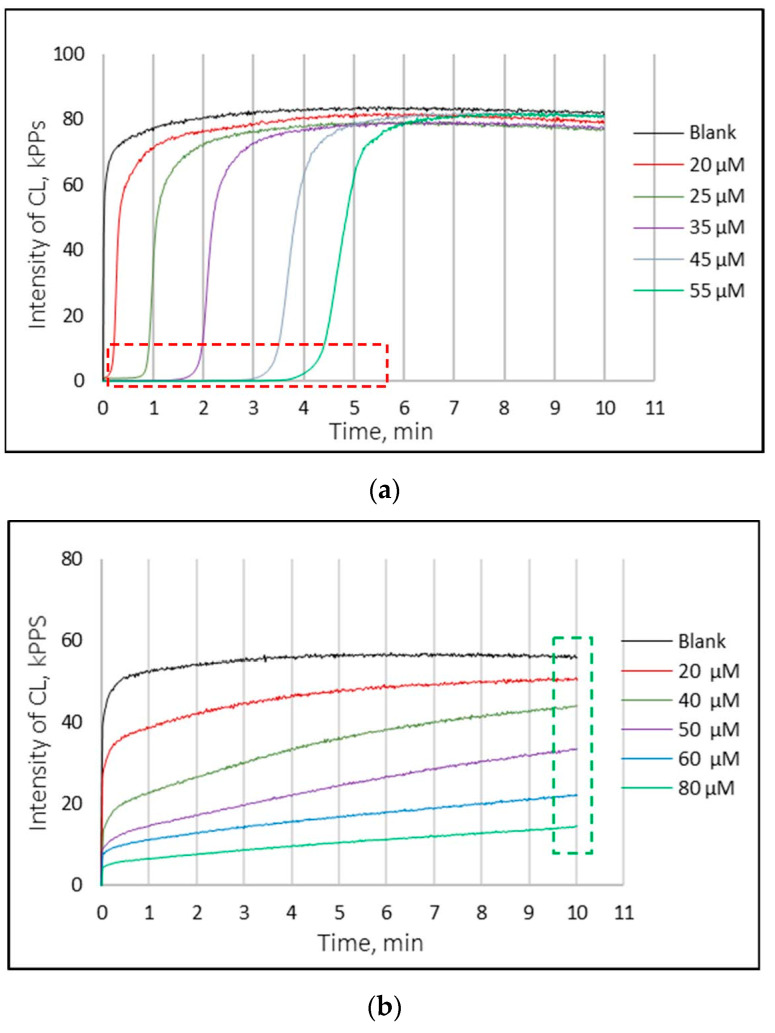
Effects of standard antioxidants on chemiluminescence kinetics: (**a**) sodium ascorbate; (**b**) tocopherol; (**c**) quercetin.

**Figure 3 antioxidants-11-01243-f003:**
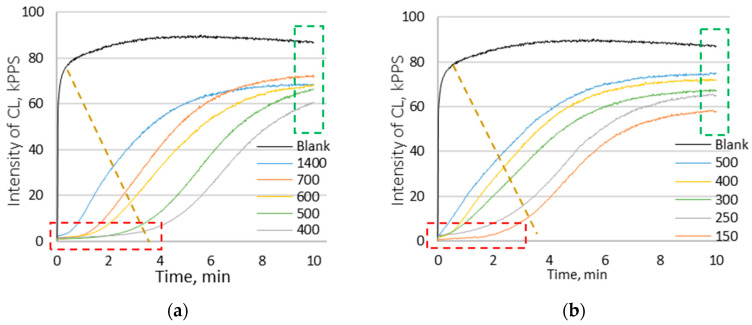
Effect of onion husk extracts on chemiluminescence kinetics: (**a**) ethanolic extract from yellow onion; (**b**) water extract from yellow onion; (**c**) ethanolic extract from red onion; (**d**) water extract from red onion; (**e**) ethanolic extract from white onion; (**f**) water extract from white onion. Different colors correspond to various dilutions.

**Table 1 antioxidants-11-01243-t001:** Compounds in the ethanolic and aqueous extracts of onion husks.

Compounds		Average (μg/g Raw Material) ± SD				
	Ethanol			Water		
	Red	Yellow	White	Red	Yellow	White
Flavonols						
Quercetin	1021.84 ± 4.07	320.86 ± 1.05 *	N.D.	83.15 ± 1.14 ^#^	10.66 ± 2.27 *^,^^	N.D.
3′-Methoxy-4′,5,7-trihydroxyflavonol	140.93 ± 2.04	12.05 ± 0.50 *	N.D.	4.85 ± 0.10 ^#^	0.37 ± 0.04 *^,^^	N.D.
Myricetin	155.84 ± 2.60	80.77 ± 1.26 *	N.D.	87.26 ± 1.05 ^#^	45.07 ± 0.23 *^,^^	N.D.
Laricitrin	16.66 ± 0.65	12.31 ± 0.15 *	N.D.	10.43 ± 0.44 ^#^	9.09 ± 0.35 *^,^^	N.D.
Flavanonols						
Taxifolin	18.70 ± 1.00	1.56 ± 0.60 *	N.D.	10.95 ± 0.18 ^#^	0.53 ± 0.11 *^,^^	N.D.
Flavonoid-O-glycosides						
Quercetin-3,4′-O-di-beta-glucoside	18.49 ± 1.66	1.50 ± 0.09 *	N.D.	3.26 ± 1.56 ^#^	0.43 ± 0.20 *^,^^	N.D.
Hyperoside	47.26 ± 0.08	0.96 ± 0.07 *	N.D.	1.06 ± 0.08 ^#^	N.D.	N.D.
Isoquercetrin	9.51 ± 0.48	0.92 ± 0.31 *	N.D.	2.61 ± 0.32 ^#^	0.34 ± 0.15 *^,^^	N.D.
Spiraeoside	485.37 ± 5.26	112.33 ± 0.59 *	N.D.	117.8 ± 0.86 ^#^	22.92 ± 0.32 *^,^^	N.D.
Isoflavones						
Tectorigenin	1.31 ± 0.59	0.78 ± 0.27	N.D.	N.D.	N.D.	N.D.
Total flavonoids	1915.90 ± 9.92	544.06 ± 2.73 *	N.D.	321.42 ± 2.61 ^#^	89.41 ± 2.08 *^,^^	N.D.

N.D.: not detected; * *p*-value ≤ 0.05 was considered significant when comparing yellow and red onion husk extracts; # *p*-value ≤ 0.05 was considered significant when comparing water with ethanolic red onion husk extracts; ^ *p*-value ≤ 0.05 was considered significant when comparing water with ethanolic yellow onion husk extracts.

**Table 2 antioxidants-11-01243-t002:** TAC of onion husk extracts determined using the FRAP and ORAC methods.

	Ethanol			Water		
	Red	Yellow	White	Red	Yellow	White
TAC_ORAC_, µmol-equiv. Trolox/g raw material	2017.34 ± 29.52 ^a^	921.47 ± 63.57 ^b,c^	16.48 ± 0.24 ^b,d^	540.02 ± 12.58 ^e,^*	342.85 ± 39.95 ^f,g,^*	5.15 ± 0.14 ^f,h^
TAC_FRAP_, µmol-equiv. DQ/g raw material	2050.23 ± 46.01 ^i^	757.61 ± 140.8 ^j,k,#^	2.03 ± 0.34 ^j,l^	709.17 ± 21.68 ^m,^*^,#^	282.63 ± 55.65 ^n,o,^*^,#^	2.73 ± 0.06 ^n,p^

^a,b,c,d^—*p*-values ≤ 0.001 were considered significant when comparing the TAC_ORAC_ of ethanolic extracts from red, yellow, and white onion husks; ^e,f,g,h^—*p*-values ≤ 0.001 were considered significant when comparing the TAC_ORAC_ of water extracts from red, yellow, and white onion husks; ^i,j,k,l^—*p*-values ≤ 0.001 were considered significant when comparing the TAC_FRAP_ of ethanolic extracts from red, yellow, and white onion husks; ^m,n,o,p^—*p*-values ≤ 0.001 were considered significant when comparing the TAC_FRAP_ of water extracts from red, yellow, and white onion husks; *—*p*-values ≤ 0.001 were considered significant when comparing the TAC of ethanolic with water extracts; ^#^—*p*-values ≤ 0.001 were considered significant when comparing TAC_ORAC_ with TAC_FRAP._

## Data Availability

Data are contained within the article or Appendix A.

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
