# Peer review of "Differences in Antioxidant Potential of Allium cepa Husk of Red, Yellow, and White Varieties"

_antioxidants, 2022, doi:10.3390/antiox11071243_

Round 1
Reviewer 1 Report
Your manuscript "antioxidants-1773793" describes the antioxidant potential of red, yellow and white onion husks extracted in 70% ethanol in comparison with water. The methods used are appropriated and well performed. Personally, I am surprised about the absence of detectable compounds in white onion husks. I strongly suggest to clarify this absence in the discussion. All sections of the manuscript are well developed. Little modifications are reported below
Page 2 lines 73-76 . in all types of onions ………… [7,9]. No detection of compounds you have found in white onions !!! why?
Page 2 lines 84-85 . Do you have the varieties of different onions?
Page 4 line 150. OHE …please indicate it.
Page 5 line 201. Table 1. Please insert : (µg/g raw material)
Page 6 lines 227-228. Please explain somewhere HAT and SET mechanism
Author Response
Dear Reviewer 1,
We would like to thank Reviewer 1 for time spent carefully reading of our article. All the given comments, which have been very helpful in improving our manuscript, have been taken into account and we present our reply to each of them separately.
Personally, I am surprised about the absence of detectable compounds in white onion husks. I strongly suggest to clarify this absence in the discussion.
Reply: We’ve added the discussion about obtained results.
Page 12 lines 423-444. Red onion husks had the highest contents of polyphenols, followed by the yellow onion husks, with the white onion containing the lowest amount [37,49,50], but in our study no compounds were detected in the white onion husk extracts. It was reported that white onions contained trace amounts of total quercetin [38], but it was detected in husk, but although in much lower concentrations than in red and yellow ones [11,14]. Quercetin and myricetin, found in large quantities in red and yellow onion husk extracts, are the main representatives of flavonols, which are considered the most abundant type of flavonoids in foods. However, the concentration of flavonols in different fruits (even those from the same species) vary due to the variation in their biosynthesis in the presence of sunlight [51]. Moreover, the antioxidant content and TAC values of onions significantly depend on the growing region [14]. Although polyphenols exist in several plant materials, however, their quantity and type are dependent on the extraction methods used, their chemical nature, the particle size, the presence of interfering compounds, and storage condition [52]. In addition, methanol has been generally found to be more efficient in extraction of lower molecular weight polyphenols [24], while we used 70% ethanol solution and water. Summarizing, we obtained the same pattern for both total flavonoid contents and TAC values: red>yellow>white. Despite there being no compounds detected in the UPLC-ESI-Q-TOF-MS analysis of the white onion husk extracts due to it extremely low concentration in prepared extract, which corresponded with low TAC values, the chemiluminescence method allowed us to evaluate the approximate ratio of antioxidants and classify them according to power.
Page 2 lines 73-76. in all types of onions ………… [7,9]. No detection of compounds you have found in white onions !!! why?
Reply: We’ve added the short explanation in the introduction.
Page 2 lines 76-78. Content of prominent flavonoids such as quercetin and its glucosides were the highest in pearls, followed by red, yellow and white onion husks [11,14] and strongly depended on extraction conditions [15].
Page 2 lines 84-85. Do you have the varieties of different onions?
Reply: We’ve studied yellow, red, and white onion husks from one harvest from Central region in Russia. We did not study the influence of environment conditions (climate, soil, season, etc.) on antioxidant potential of selected onion varieties. But we’re very grateful to Reviewer for the idea, because we’ve started to collect such information for yellow onion husk.
Page 4 line 150. OHE …please indicate it.
Reply: OHE is onion husk extracts, the abbreviation was changed into onion husk extracts (Page 4 lines 152-153).
Page 5 line 201. Table 1. Please insert: (μg/g raw material)
Reply: In was inserted (Page 5 line 203).
Page 6 lines 227-228. Please explain somewhere HAT and SET mechanism
Reply: The explanation was added in Section 3.2.
Page 6 lines 228-233. The ORAC fluorescent method was used to evaluate the contribution of antioxidants demonstrating an HAT (hydrogen atom transfer) mechanism of action and describing the ability of antioxidants to neutralise free radicals by hydrogen donation. The FRAP photometric method was used to evaluate the contribution of antioxidants demonstrating an SET (single electron transfer) mechanism of action and describing the ability of antioxidants to interact with free radicals by transferring a single electron [22].
Best regards,
Authors.
Reviewer 2 Report
This is a well-organized and nicely written manuscript. Although the amount of bio-waste generated from the onion industry is less than 1% (based on figures provided by authors), it still constitutes an environmental burden and a resource worthy of exploitation within the circular economy model. One minor observation is the lack of any mention of sulfur compounds that may have been detected among the 100 detected by LC/MS. Any comment on this would enrich the quality of this paper. Otherwise, it was an enjoyable read.
Author Response
Dear Reviewer 2,
We would like to thank Reviewer 2 for time spent carefully reading of our article. The given comment, which have been very helpful in improving our manuscript, have been taken into account and we present our reply to this comment.
One minor observation is the lack of any mention of sulfur compounds that may have been detected among the 100 detected by LC/MS.
Reply: According to Table S1 (Supplementary Materials), lipoic acid was also identified in extracts, which belonged to organosulfur acids. Therefore, we’ve rewritten the following sentence:
Page 4 lines 194-195. «….including organic (n=3), organosulfur (n=1)…..»
However, the presence of flavonoid standards does not allow to evaluate lipoic acid content, because of a different structure of the compound. Hence, structure of the lipoic acid will demonstrate a different ionization and response of lipoic acid ions on the mass spectrometer. Therefore, there may be large errors in the interpretation of the data of the quantitative characteristic of lipoic acid. Thus, we’ve described and discussed only reliable data about quantitative characteristics of flavonoids.
Best regards,
Authors.